# Longshot enables accurate variant calling in diploid genomes from single-molecule long read sequencing

Peter Edge[1] & Vikas Bansal[2]*

Whole-genome sequencing using sequencing technologies such as Illumina enables the accurate detection of small-scale variants but provides limited information about haplotypes and variants in repetitive regions of the human genome. Single-molecule sequencing (SMS) technologies such as Pacific Biosciences and Oxford Nanopore generate long reads that can potentially address the limitations of short-read sequencing. However, the high error rate of SMS reads makes it challenging to detect small-scale variants in diploid genomes. We introduce a variant calling method, Longshot, which leverages the haplotype information present in SMS reads to accurately detect and phase single-nucleotide variants (SNVs) in diploid genomes. We demonstrate that Longshot achieves very high accuracy for SNV detection using whole-genome Pacific Biosciences data, outperforms existing variant calling methods, and enables variant detection in duplicated regions of the genome that cannot be mapped using short reads.

---

[1] Department of Computer Science and Engineering, University of California, San Diego, La Jolla, California 92093, USA. [2] Department of Pediatrics, School of Medicine, University of California, San Diego, La Jolla, California 92093, USA. *email: vibansal@ucsd.edu

The availability of second-generation DNA sequencing technologies such as Illumina short reads has made the resequencing of human genomes routine[1]. Both single-nucleotide variants (SNVs), the most abundant form of variation in the human genome, and small indel variants can be reliably detected using whole-genome Illumina sequencing using sequence coverage of 30–40×[2,3]. Nevertheless, sequencing human genomes using short-read sequencing technologies has many limitations. First, humans are diploid organisms with two copies (maternal and paternal) of each autosomal chromosome. Haplotypes, or the sequence of alleles that occur on an individual chromosome, can be computationally assembled from whole-genome sequencing (WGS) using overlaps between reads that span multiple heterozygous variants[4–6]. However, due to the low rate of heterozygosity of human genomes[7], Illumina reads derived from paired-end sequencing of short fragment libraries (200–500 bp in length) typically cover only a single variant site and do not provide long-range haplotype information. Second, ~3.6% of the genome consists of long and highly similar duplicated sequences where short reads cannot be uniquely mapped and hence SNVs cannot be detected. These regions overlap hundreds of coding genes, including many disease-associated genes such as *PMS2* and *STRC*[8].

Third-generation single-molecule sequencing (SMS) technologies such as Pacific Biosciences (PacBio) and Oxford Nanopore Technologies (ONT) generate long sequence reads; average read lengths for the PacBio single-molecule real-time (SMRT) technology are 10–30 kb[9]. These long reads have the potential to overcome many of the limitations of short-read sequencing technologies including haplotyping and detection of structural variation. Indeed, SMS data have been successfully used for de novo assembly of human genomes[10,11], identifying complex structural variation[12] and haplotype assembly of human genomes[10,13]. However, compared with short-read sequencing technologies such as Illumina, the per-base accuracy of SMS reads is low with an error rate exceeding 10% (primarily due to insertion/deletion errors)[9]. This high error rate makes the detection of small sequence variants such as SNVs, particularly heterozygous variants, difficult.

With the decreasing cost of SMS technologies and their increasing use for sequencing human genomes, accurate short variant calling methods for long-read SMS data can be valuable in many ways. Current benchmarks for variant calling in human genomes, developed by the the Genome in a Bottle (GIAB) Consortium[14,15], are based on short-read sequence data and cover ~90.8% of the reference human genome sequence. These high-confidence variant calls are immensely valuable for developing new variant calling methods and sequencing technologies. However, these variant call sets are biased towards regions of the genome that are easy-to-call using short reads[16]. Accurate SNV calling using long-read SMS data can provide independent validation of short-read SNV calls leading to reduction in false positives and increased understanding of systematic errors and artifacts. Furthermore, SNV calling using SMS reads can enable the generation of high-confidence variant calls in repetitive regions of the genome that include segmental duplications. The ability to call variants in repetitive regions that are inaccessible to short-read sequencing technologies can also advance the use of SMS technologies for detection of disease-causing mutations in duplicated genes via whole-genome or targeted sequencing[17].

Haplotype-resolved SNV detection from SMS reads can also enable the discovery of other types of human genetic variation, such as structural variants (SVs) via separation of reads using haplotypes. Huddleston et al.[18] used an assembly-based approach, SMRT-SV, to identify thousands of SVs from whole-genome PacBio data of two haploid genomes, 89% of which were not reported by the 1000 Genomes Project[19]. However, the sensitivity of SV detection using SMRT-SV was only 41% in diploid genomes. Chaisson et al.[20] performed dense whole-genome haplotyping of a human genome using multiple sequencing technologies, and were able to call SVs successfully on each group of haplotype-separated SMS reads.

Variant calling tools such as GATK HaplotyperCaller[21] and FreeBayes[22] developed for short-read data analysis are not well-suited for SNV detection using PacBio data for two reasons as follows: (i) short reads have low error rates (<0.5%) and these methods do not model the high indel error rate of SMS reads, which makes it difficult to distinguish true SNVs from errors, and (ii) these methods analyze reads in short windows (typically a few hundred bases) and are not designed to leverage the haplotype information present in SMS reads. This haplotype information can be invaluable in distinguishing true variants from errors, as observations of a true variant segregate with the reads originating from the haplotype on which it occurs, whereas sequencing errors are unlikely to segregate. Recently, several methods for variant calling from long reads and deep-learning-based variant calling methods have been developed[23–25]. However, the accuracy of these methods for SNV calling on SMS data is currently much lower than that using Illumina WGS[24,25].

We describe a diploid SNV calling method, Longshot, which harnesses long SMS reads to jointly perform SNV detection and haplotyping. For this, it uses our read-based haplotype phasing method HapCUT2[13]. To overcome the high error rate of SMS reads, it utilizes a pair-Hidden Markov Model (pair-HMM) to average over the uncertainty in the local alignments and estimate accurate base quality values that can be used for calculating genotype likelihoods. We benchmarked Longshot using simulated data and whole-genome SMS data for multiple human individuals sequenced using the PacBio SMRT and Oxford Nanopore sequencing technologies[14,15,26]. LongShot achieves very high accuracy for SNV detection (precision ≥ 0.992 and recall ≥ 0.96) on PacBio SMS datasets and outperforms current variant calling methods in accuracy and run time. We find that Longshot can also call SNVs with high accuracy using whole-genome Oxford Nanopore data.

## Results

**Overview of the method.** Alignments of SMS reads suffer from reference bias, which can cause an SNV allele to be obscured by gaps (insertions and deletions) in the alignments (Supplementary Fig. 1). Nevertheless, a true SNV is likely to have at least a few correctly aligned reads with the alternate allele. The first step in the Longshot algorithm identifies potential SNV sites using a standard pileup-based genotyping calculation[27] (Fig. 1a). A low variant quality threshold is used to select SNVs to minimize false negatives. Next, for each candidate SNV, we determine the most likely allele for each read covering the SNV and the corresponding estimate of the quality of the allele call (Fig. 1b). This allelotyping is done by local realignment of a segment of the read to short haplotype sequences (one for each of the two alleles at a biallelic SNV site). In low-complexity regions of the genome (e.g., homopolymers), there is significant ambiguity in the placement of gaps for SMS reads and many alignments are equally likely[28]. Therefore, we use the forward algorithm on a sequence alignment pair-HMM[29] to perform the local realignment by averaging all possible local alignments of a read to a given haplotype.

After estimating the allele call and quality value for each read overlapping an SNV site, we estimate phased genotypes for all SNVs simultaneously using a haplotype-based likelihood model (see Methods). SMS reads typically cover multiple heterozygous sites and this haplotype information is useful, as an SNV on a

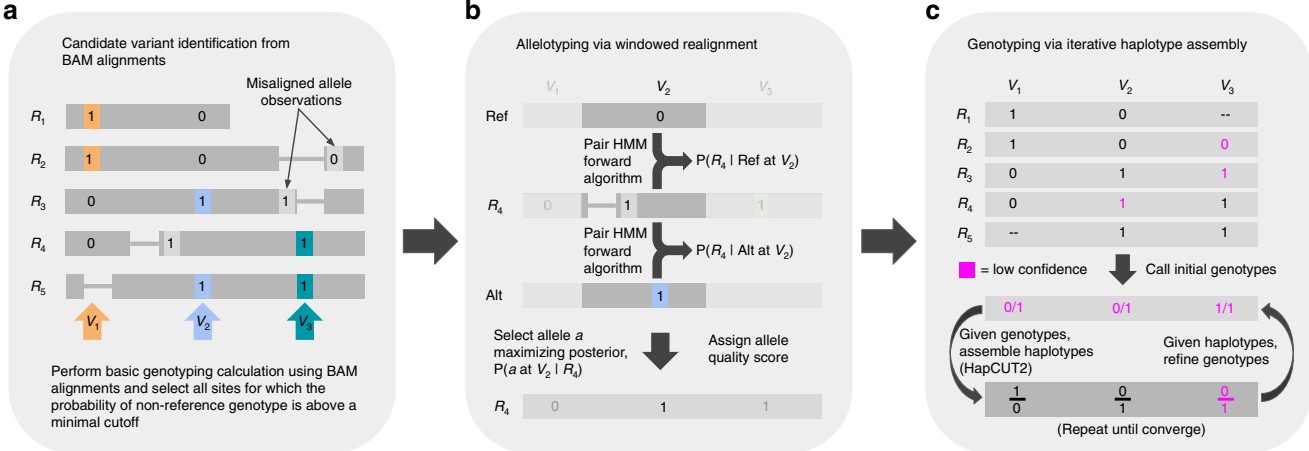

**Fig. 1** Overview of the Longshot algorithm. **a** Candidate variants are identified using the pileup of the original alignments and a standard genotype likelihood calculation is used to determine whether the site is a potential variant. **b** To determine the allele for each read at each potential SNV site it overlaps, a window is formed around the variant and the probability of the observed read sequence given each allele is calculated using the forward algorithm on a Pair-Hidden Markov Model. The most likely allele and quality score is chosen based on the relative likelihoods of the two alleles. **c** Using the alleles and quality values for each read at variant sites, phased genotypes for all variants are determined jointly by performing haplotype assembly using HapCUT2 (on heterozygous variants) and local updates of the phased genotypes in an iterative manner

haplotype is expected to segregate with reads from the same haplotype (although random sequencing errors are not). In Longshot, heterozygous SNVs are assembled into haplotypes using HapCUT2 and a local update procedure is used to estimate the most likely phased genotype for each SNV given the current haplotypes for all other SNVs (Fig. 1c). This procedure is repeated for a few iterations until the likelihood stops improving. Finally, the variants are filtered for maximum read coverage, excessive variant density, and minimum Genotype Quality (GQ) score, where the GQ score is estimated using the phased genotype likelihoods.

**Accurate SNV calling using simulated data**. First, we used simulations to assess the accuracy of SNV calling using Longshot and also compared the precision and recall with short-read variant calling. We simulated a diploid genome by adding SNVs to the reference human genome and simulated paired-end Illumina reads and PacBio SMS reads from this genome (maximum coverage of 60×). Subsequently, we aligned the reads to the reference genome using BWA-MEM[30] (Illumina) and BLASR[31] (PacBio), and called SNVs using FreeBayes and Longshot, respectively. Across the entire genome, the precision was consistently high (≥0.9999) at all read coverages (20–60×) for both short read and SMS read-based SNV calling (Supplementary Fig. 2). Short reads achieved greater recall than SMS reads at lower coverage (≤30×), whereas SMS reads had marginally greater recall at higher coverage (≥40×). SMS reads are expected to have better mappability in repetitive regions of the genome compared with Illumina reads, particularly in long segmental duplications with high sequence identity. Indeed, the recall for SMS reads in segmental duplications with high sequence similarity (≥95%, 127.5 Mb of DNA sequence) was significantly higher (0.86 at 40× coverage) compared with that using short reads (0.57 at 40× coverage) and increased with increasing coverage (Supplementary Fig. 2).

We also compared the precision/recall of SNV calling using BLASR with several long-read mapping tools: NGMLR[32], BWA-MEM[30], and MINIMAP2[33]. All tools showed high precision and recall when considering SNVs across the whole genome, but BLASR had significantly higher recall (maximum of 0.88) than all other aligners (0.72 using Minimap2) in segmental duplications

(Supplementary Fig. 2). Therefore, we utilized BLASR for the analysis of real datasets.

We used the simulated datasets to estimate the theoretical fraction of the genome that is callable with SMS long reads compared with short reads at 60× coverage. We found that SMS reads were able to span 99.4% of the genome (non-N bases on chromosomes 1–22 with at least 30× coverage and at least 90% of reads well-mapped at each position (Supplementary Fig. 3)). In comparison, Illumina reads covered 96.3% of the genome under these same criteria, a difference of 3.1%.

**Accurate SNV calling using whole-genome PacBio data**. We used Longshot to call SNVs using whole-genome human PacBio data for four human genomes from GIAB consortium[14]. Specifically, we used WGS data for the NA12878 individual (45×) and a mother–father–child trio of Ashkenazi ancestry (NA24385 at 64×, NA24149 at 29×, and NA24143 at 27×). For each dataset, a GQ threshold that was linearly proportional to the median read depth was used for filtering variants (see Methods). For comparison, we also called SNVs using Illumina short-read WGS data (~30× coverage) for each individual.

Longshot identified 3.51 to 3.65 million SNVs per genome (on chromosomes 1–22 only) and required 35 h on average to process ~28× whole-genome data on a single core (Supplementary Table 1). To assess the precision and recall of SNV calling, we utilized the GIAB high-confidence variant call set for each individual[14,15]. The comparison of SNV calls was limited to GIAB high-confidence regions for each genome[15]. The precision and recall for NA12878 were 0.9942 and 0.9592, respectively, at 30× coverage and the recall improved to 0.9734 at 45× coverage. The precision and recall, and the precision-recall curves (Supplementary Fig. 4) were highly consistent across the four genomes at 27–30 coverage (Fig. 2a,b), demonstrating the robustness of our method. To assess the improvement in precision/recall as a function of sequence coverage, we sub-sampled data for the AJ son individual (NA24385), who was sequenced to 64× coverage. The recall improved steadily from 0.9608 (28×) to 09798 (64×), whereas the precision only changed moderately with increasing coverage (0.9930 to 0.9936). The precision and recall for SNV calling using SMS reads was slightly lower than Illumina-based variant calling (Fig. 2). Nevertheless, the ability of Longshot to

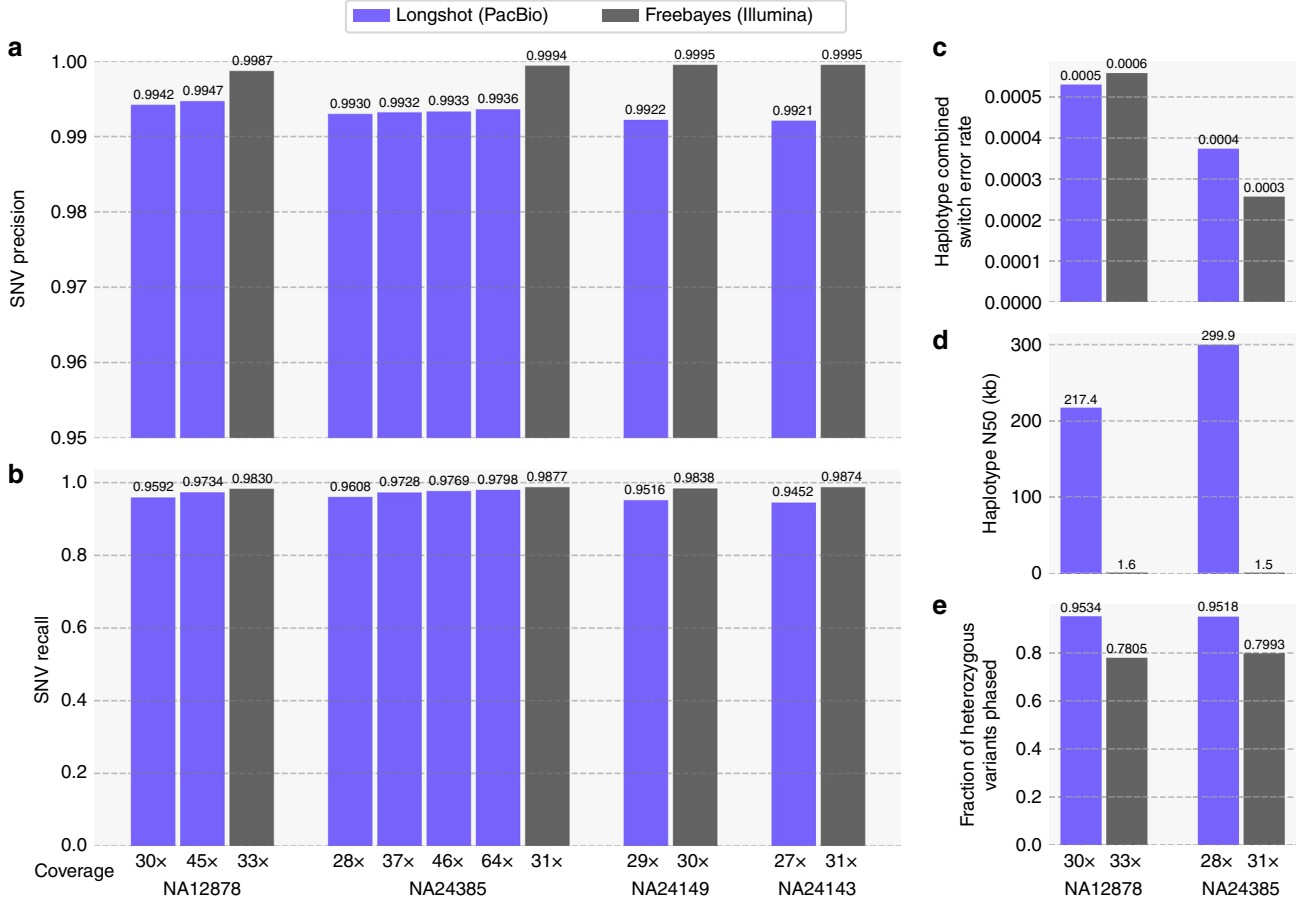

**Fig. 2** Accuracy and completeness of LongShot SNV calls on whole-genome SMS data. Longshot was used to call single nucleotide variants (SNVs) using SMS data from the GIAB project for four human genomes: NA12878 (30× and 45× coverage), NA24385 (28×, 37×, 46×, and 64× coverage), NA24149 (29× coverage), and NA24143 (27× coverage). For each individual, variants were also called using FreeBayes applied to ∼30× coverage Illumina short reads. **a** Precision of the SNV calls calculated using the GIAB high-confidence variant call set. **b** Recall of the SNV calls. **c** The combined switch error rate (total rate of switch errors and mismatch errors) of the Longshot and Illumina short-read-based haplotypes. **d** N50 length of the haplotypes. **e** The fraction of heterozygous variants phased in each dataset

consistently achieve high recall (only 2–3% lower than Illumina WGS for the same depth of coverage), while achieving a low false discovery rate (FDR; average = 0.7%) was remarkable given the significantly high error rate of SMS reads (∼10%) compared with Illumina reads.

In contrast with simulated data, the precision of of Longshot on real SMS reads was slightly lower than short-read variant calling. To determine the source of false-positive calls, we analyzed whether such calls were enriched in specific sequence contexts or overlapped with indels. For the NA12878 dataset (Supplementary Table 2), we observed that the vast majority (71.4%) of false-positive SNVs are located within 5 bp of a true indel. These false-positive SNVs are called, as the current implementation of Longshot does not consider indels as potential variants. Filtering SNV calls located near known indels (using the Mills + 1000 Genomes Gold Standard Indels set from the GATK resource bundle[21]), reduced the number of false positives by 34–45% for the four GIAB genomes (Supplementary Table 3), while only slightly decreasing the recall. Analysis of false-negative SNVs showed that 19.5% of the false-negative SNVs occurred inside homopolymer sequences of length 5 or greater, which is 3.4× the expected value. This follows naturally from the fact that these regions have low information content; insertion and deletion errors could plausibly lie anywhere along the length of a homopolymer. Therefore, allele calls inside homopolymers

receive lower quality scores from the pair-HMM realignment, which reduces the power to call SNVs in such regions.

To compare Longshot's accuracy on SMS data with other methods, we considered existing variant calling methods for short-read data including GATK and FreeBayes. However, the GATK HC tool did not generate variant calls on the NA12878 PacBio dataset, consistent with previous evaluations of these methods on SMS data[24]. Recently, a deep-learning-based method for variant calling has been developed that can process both Illumina and SMS long-read data[24]. Although we were unable to perform a direct comparison with DeepVariant due to unavailability of trained models for PacBio continuous long read (CLR) data, comparison of the reported precision and recall for DeepVariant on the NA12878 dataset (aligned with the same tool) showed that Longshot had better precision than DeepVariant, while the recall was similar (Supplementary Table 4). At a GQ cutoff of 36, Longshot had the same recall as DeepVariant but higher precision (0.9939 vs. 0.9819).

We directly compared the accuracy of Longshot with a deep-learning-based method Clairvoyante[34] and WhatsHap[25], using whole-genome SMS data for four individual genomes. We used reads aligned with the NGMLR aligner[32] for evaluation, as Clairvoyante provides trained models for this aligner (see Supplementary Methods for details). For WhatsHap, we used the potential variants identified in step 1 of Longshot as input, as

**Table 1 Comparison of accuracy for variant calling methods on whole-genome SMS data**

| Genome | Read Coverage | Method | Precision | Recall | Runtime (h) |
|--------|---------------|--------|-----------|--------|-------------|
| NA12878 | 44 | Longshot | 0.995 | 0.968 | 23:31 |
| | | WhatsHap | 0.972 | 0.975 | 27:47 |
| | | Clairvoyante | 0.984 | 0.957 | 21:44 (×4) |
| NA24385 | 62 | Longshot | 0.987 | 0.965 | 41:55 |
| | | WhatsHap | 0.976 | 0.974 | 32:09 |
| | | Clairvoyante | 0.990 | 0.969 | 22:25 (×4) |
| NA24385 | 27 | Longshot | 0.981 | 0.927 | 20:03 |
| | | WhatsHap | 0.959 | 0.941 | 22:54 |
| | | Clairvoyante | 0.960 | 0.927 | 21:09 (×4) |
| NA24143 | 27 | Longshot | 0.993 | 0.941 | 18:51 |
| | | WhatsHap | 0.962 | 0.949 | 22:06 |
| | | Clairvoyante | 0.960 | 0.920 | 21:42 (×4) |
| NA24149 | 23 | Longshot | 0.993 | 0.924 | 16:59 |
| | | WhatsHap | 0.959 | 0.934 | 20:30 |
| | | Clairvoyante | 0.938 | 0.904 | 23:59 (×4) |

All methods were run on BAM files generated using the NGMLR aligner, and precision and recall values were calculated using the GIAB high-confidence variant calls. The runtime listed is the total walltime to process all chromosomes individually. Clairvoyante supports multi-threading and was run using four threads per chromosome

the current version of this tool (version 0.18) does not support potential variant identification. On the NA12878 dataset, the precision and recall for Longshot were higher than both Clairvoyante and WhatsHap (Table 1). In particular, Longshot achieved very high precision or a low FDR of 0.5%. In comparison, the FDR for Clairvoyante was threefold higher, 1.6%. Comparison of the precision-recall curves for three methods on the NA12878 dataset showed that Longshot outperforms both competing methods for all precision values >0.98 (Supplementary Fig. 5). Similarly, analysis of variant calls for two other GIAB genomes (NA24143 and NA24149) showed that Longshot had the best precision and recall among the three methods (1). On the high-coverage NA24385, Clairvoyante's recall and precision were marginally better than Longshot (0.3–0.4% higher). Nevertheless, the precision (0.994) and recall (0.980) for Longshot on this dataset using the BLASR alignments were better than Clairvoyante (0.990 and 0.969, respectively). Longshot was also the most computationally efficient of three methods in terms of run time (1). For the NA12878 dataset, the maximum memory usage for Longshot was 5.5 GB compared with 6.2 and 12.7 GB for WhatsHap and Clairvoyante, respectively.

The phased genotyping or haplotype assembly step of Longshot distinguishes it from state-of-the-art variant callers for short-read data[21,22] and recent deep-learning-based methods for variant calling[24,34]. We investigated the importance of the phased genotyping for the accuracy of Longshot by running it on the NA12878 PacBio dataset (downsampled to 30× coverage) without phased genotyping (essentially skipping step 3 of the algorithm). We found that skipping the phased genotyping reduced Longshot's recall significantly from 0.959 to 0.905 (GQ threshold of 30), while the precision remained virtually unchanged (Supplementary Fig. 6).

**Accuracy of Longshot haplotypes**. Next, we assessed the accuracy and completeness of haplotype assembly using Longshot for two GIAB individuals, NA12878 and NA24385, by comparison with gold-standard haplotypes for these individuals inferred using pedigree data (see Methods). The median read lengths for these two datasets were 3587 and 7235 bp, respectively. The Longshot haplotypes for NA12878 had an N50 length of 217.4 kb (with respect to the phased portion of the genome) and were very

accurate, with a combined switch error rate of 0.05% (Fig. 2c,d). Similarly, the haplotypes for NA24385 (30× coverage) had an N50 length of 299.9 kb and a combined switch error rate equal to 0.04%. In comparison, haplotypes assembled using short reads had a N50 length <2 kb for both genomes (Fig. 2d). We also used HapCUT2 and WhatsHap to assemble haplotypes for NA12878 and NA24385 using SMS reads and SNVs identified using ~ 30× coverage Illumina sequencing[13]. We found that the haplotype accuracy and completeness were comparable between the three methods, whereas HapCUT2 had the lowest switch and mismatch error rates (Supplementary Fig. 7). Separation of SMS reads using SNV haplotypes can enable discovery of non-SNV variants such as indels and SVs using methods such as SMRT-SV[11], which work well on haploid genomes. For the NA12878 dataset (chromosome 1 only), 51.1% of reads (weighted by length) could be assigned to a haplotype with high confidence. The ability to assign reads to haplotypes was dependent on read length: the haplotype-assigned reads had a median length of 4.3 kb, whereas the unassigned reads had a median length of 2.6 kb only.

**SNV calling using Oxford Nanopore reads**. Recently, reads from Oxford Nanopore Technologies' (ONT) MinION sequencer were used to assemble a human genome[26]. Nanopore reads have a similar error profile to PacBio SMRT reads; however, the total per-base error rate of ONT reads is reported to be higher than for PacBio SMRT[35] and the errors are dependent on sequence context[36]. We applied the Longshot algorithm to call SNVs using a whole-genome Oxford Nanopore dataset for a human individual (NA12878, 37× coverage). We observed that the candidate set of SNVs considered by Longshot contained a significant fraction of false positives due to the context-specific errors in Nanopore reads. To ameliorate this, we implemented a simple filter to remove potential SNVs for which the allele observations show a significant strand bias (Fisher's exact test $p$-value < 0.01), prior to haplotype assembly. On the latest version of this ONT dataset, LongShot achieved a precision equal to 0.991 and recall value equal to 0.933 at a GQ threshold of 65 for SNV calling (see Supplementary Fig. 8 for a precision-recall curve). For comparison, we called variants using Nanopolish, a software tool for signal-level analysis of Oxford Nanopore data[36]. Nanopolish required more than 43 h to call variants on chromosome 20 using 4 cores and achieved a best F1 score of 0.93 (Supplementary Fig. 8). In contrast, Longshot had a best F1 score of 0.967 and took only 5 h and 13 min for variant calling (using a single core). In addition, the accuracy of Longshot on Oxford Nanopore data was better than the reported accuracy of other methods (Supplementary Table 4).

**Analysis of SNV calls in repetitive regions**. As demonstrated with simulations, the recall of variant calling using SMS reads in segmental duplications with high sequence similarity (≥95%, Fig. 2) is significantly higher (0.86) compared with short reads (0.57). These regions correspond to 102.8 Mb of the genome (excluding the sex chromosomes). However, 97.7% of these regions are excluded from the GIAB high-confidence variants, making it challenging to assess the accuracy of SNV calling using real SMS data. We compared SNV calls in segmental duplications for the NA12878 genome made using short-read Illumina data (33 × coverage) and SMS reads (30× coverage). In segmental duplications with ≥95% similarity, 180, 889 SNVs were called using SMS reads, 55.0% more than those using Illumina reads (Table 2). The fewer calls using Illumina reads likely reflect the inability to map in segmental duplications. For example, Illumina reads cannot be mapped uniquely in a significant portion of the *STRC* gene, resulting in 52.3% fewer variants called compared

**Table 2 Comparison of PacBio and Illumina SNV calls for NA12878**

|  |  | Genome (1–22) | Inside GIAB Confident | Outside GIAB Confident | Segmental Dup. (≥95% similar) | Segmental Dup. (≥99% similar) |
|---|---|---|---|---|---|---|
| Region size |  | 2.8 Gb | 2.4 Gb | 330.7 Mb | 102.8 Mb | 47.5 Mb |
|  | # SNVs | 3,518,530 | 3,002,660 | 515,870 | 180,889 | 78,851 |
| PacBio | Ts/Tv | 2.08 | 2.14 | 1.75 | 1.95 | 1.99 |
|  | # SNVs | 3,563,787 | 3,065,573 | 498,214 | 116,649 | 18,684 |
| Illumina | Ts/Tv | 2.03 | 2.1 | 1.66 | 1.84 | 1.79 |
|  | # SNVs | 254,428 | 63,848 | 190,580 | 103,621 | 69,705 |
| Unique to PacBio | Ts/Tv | 1.63 | 1.83 | 1.57 | 1.9 | 1.99 |
|  | # SNVs | 299,733 | 126,763 | 172,970 | 39,409 | 9538 |
| Unique to Illumina | Ts/Tv | 1.3 | 1.26 | 1.33 | 1.53 | 1.55 |
| Shared | # SNVs | 3,264,078 | 2,938,812 | 325,266 | 77,241 | 9146 |
| Illumina & PacBio | Ts/Tv | 2.12 | 2.15 | 1.85 | 2.01 | 2.04 |

Variants were called using short reads (33× coverage) with FreeBayes and using SMS long reads (30× coverage) with Longshot. The number of variants called by each technology, the number of variants shared between the two technologies, and the corresponding transition/transversion (Ts/Tv) ratios are shown for the whole genome and various subsets of the genome including GIAB high-confidence regions and segmental duplications with high sequence identity

with SMS reads (Fig. 3). We found that in total, 1.66 Mb of the bases in segmental duplications with ≥95% similarity overlap with coding exons and 90.3% of these bases were well-mapped in the 45× PacBio dataset (each position having at least 20× coverage and ≥90% of reads aligned to the position having MAPQ ≥30). The difference was more stark in segmental duplications with ≥99% similarity: 78, 851 SNVs were called with SMS reads compared with only 18, 684 with Illumina reads (4.2-fold difference). The Transition/Transversion (Ts/Tv) ratio for the SNVs called only using SMS reads in these regions was 1.99, slightly lower than the ratio for the SNV calls in GIAB confident regions (~2.1). This is consistent with the expectation that the Ts/Tv ratio is usually ~2.0–2.1 for SNVs across the whole genome[37]. In contrast, the Ts/Tv ratio for Illumina-only calls in segmental duplications with ≥99% similarity was 1.55, much lower than the expected value (Table 2).

Next, we assessed the Mendelian consistency of SNV calls for the mother–father–child trio of Ashkenazi ancestry from the GIAB project. To minimize discordance due to false negative calls, only sites with at least 20× read coverage in every individual were considered. SMS calls in the high-confidence GIAB regions had higher concordancy (98.88%) compared with calls outside GIAB confident regions (96.17%). Within segmental duplications (≥95% similarity), 4.99% of the SNVs in the child were discordant with Mendelian inheritance. Many of the discordant SNVs were clustered in contiguous blocks, indicating that they are the result of mismatched reads or structural variation in one or more individuals.

Finally, we compared Longshot SNV calls for NA12878 to the Platinum Genomes small variant call set for this genome that have been generated using Illumina WGS and validated using haplotype inheritance on a 17-member pedigree[38]. In GIAB high-confidence regions, 95.2% of the PG SNVs were also called by Longshot. The PG calls cover a significant fraction of the genome (330.7 Mb) that is excluded from the GIAB high-confidence calls. In these regions, only 79.6% of the PG SNVs were shared with Longshot and 74,641 SNVs were unique to the PG calls (Supplementary Fig. 10). The low concordance in regions outside the GIAB high-confidence regions highlights the challenge of accurate variant calling in these regions. Longshot's ability to call SNVs accurately using SMS reads provides an orthogonal validation for SNVs called using short reads. In-depth analysis of variant calls made using short-read and SMS data in these regions can enable the expansion of confidently called regions for reference human genomes.

## Discussion

Our results demonstrate that highly accurate detection of SNVs is feasible even from long-read sequence data with high error rates. Combined with recent work demonstrating the ability to detect and genotype SVs from SMRT-seq data, our results indicate that long-read sequencing can be used to accurately detect all forms of genetic variation in human genomes. Recently, Li et al.[16] wrote that "although PacBio assembly is accurate at the base-pair level for haploid genomes, it is currently not accurate enough to confidently call heterozygotes in diploid mammalian genomes." We have demonstrated that heterozygous SNVs can be called accurately in diploid genomes, by combining sensitive allelotyping of reads at SNV sites with haplotype-informed genotyping. Our method has a very low FDR (0.5–0.8%) across multiple whole-genome PacBio datasets that is two- to fourfold lower than other variant calling methods. Furthermore, we find that the FDR can be reduced further to 0.3% by filtering out known common indels.

We have also demonstrated that SMS reads can be used to call SNVs in segmental duplications and other regions of the genome with low short-read mappability. However, correctly mapping PacBio reads in highly similar segmental duplications remains a challenge. As Supplementary Fig. 2 shows, there is a wide variance in the ability of SMS read mappers to map reads in segmental duplications. This is likely due to the mappers having different strategies for dealing with highly similar mappings that are differentiated by a small number of paralog-specific variants. Despite BLASR performing relatively well using simulated reads, many of the discordant SNVs observed between the AJ trio in segmental duplications appeared to be caused by the presence of multiple mismapped reads. SMS read mapping methods with specific optimizations for segmental duplications could improve the ability to call variants in segmental duplications[33].

The GIAB and Platinum Genomes variant sets used to assess variant-calling accuracy in this study were generated using short-read datasets, are therefore biased in favor of short-read technologies[16], and exclude regions where long reads are likely to have better precision and recall. Therefore, in an unbiased genome-wide comparison, Longshot may achieve even better accuracy than short-read variant calling methods. Furthermore, some of the false-positive calls by Longshot may actually correspond to false negatives in the GIAB high-confidence call sets. A recent graph-based read alignment approach identified thousands of variants that were absent in the GIAB call sets[39]. In the NA12878 genome, Longshot identified 5900 SNVs that are

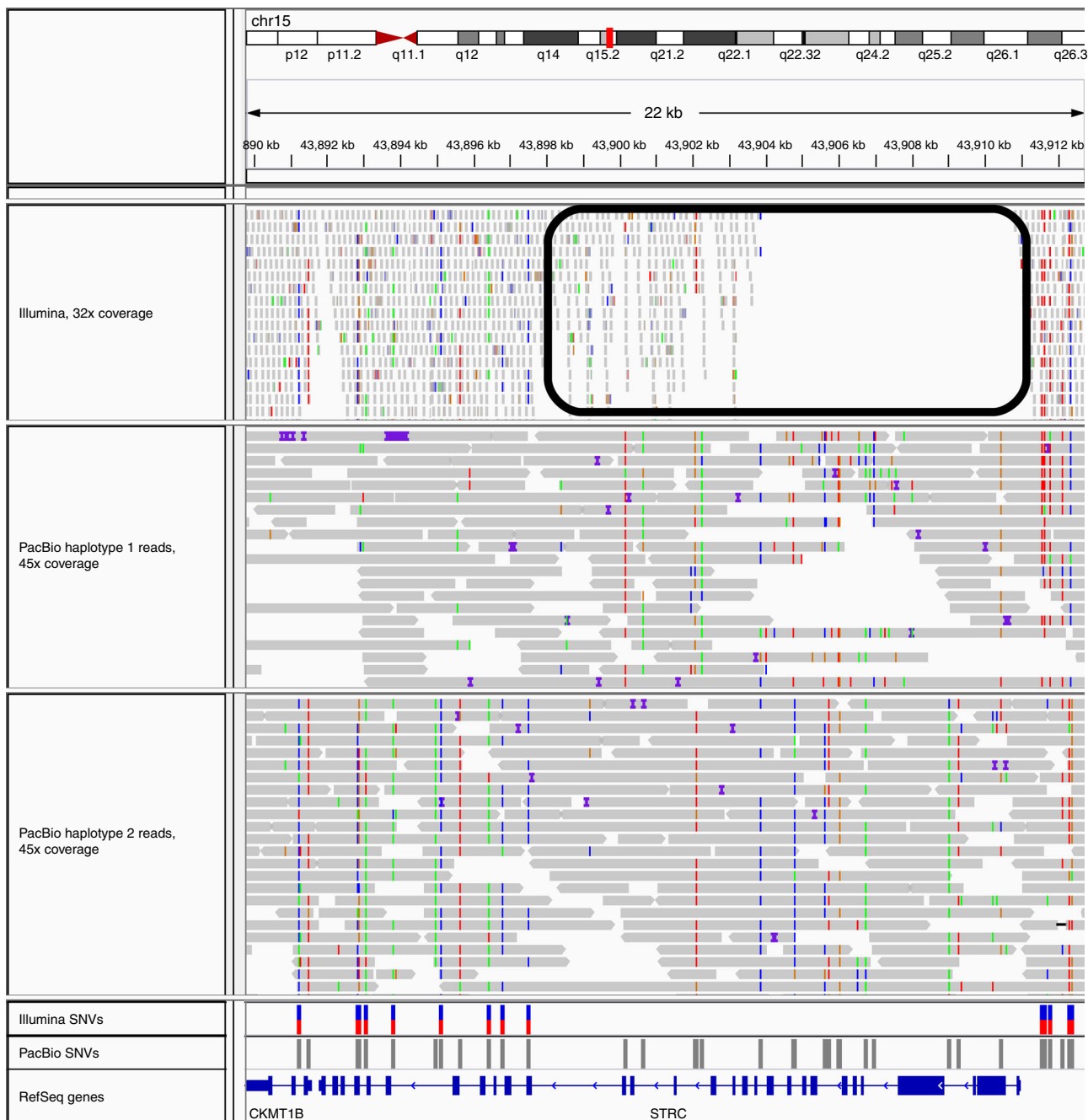

**Fig. 3** Accurate variant calling using SMS reads and Longshot in the duplicated gene *STRC*. An Integrated Genomics Viewer (IGV) view of mapped reads shows that a long segment of the gene (circled in black) has low coverage using uniquely mapped Illumina reads due to the presence of a long segmental duplication with high sequence similarity (> 99.8%) that spans the entire gene. PacBio reads (separated by haplotype using Longshot phased SNVs) have consistent coverage of mapped reads across the entire gene, allowing Longshot to call 42 SNVs of which 20 are shared with short reads, and 22 are unique to Longshot

located in GIAB high-confidence regions and do not overlap indels present in the GIAB variant calls. Many of these variants are located in variant-dense genomic regions that are problematic for mapping using short reads but should be callable using long single-molecule reads. Further analysis of these variants will be helpful in improving the recall of gold-standard variant call sets for human genomes.

Longshot offers the ability to assemble haplotypes without prior knowledge of SNVs and leverage the haplotypes to separate SMS reads by haplotype. This opens up a wide range of

possibilities for SMS read analysis, given that many SMS analysis tools work much better on haploid samples. For example, the haplotype-separated reads could be used to call SVs with greater sensitivity using a tool such as SMRT-SV[11]. A similar approach was recently used to profile structural variation genome-wide after extensive haplotype assembly with multiple sequencing technologies and computational separation of the SMS reads by haplotype[20]. Currently, Longshot uses the read pileups to identify candidate SNVs and the vast majority (~72%) of false-positive SNVs identified with Longshot correspond to misclassified indel

variants. Using a genomic consensus of haplotype-separated reads should improve the accuracy of variant calling using Longshot.

LongShot was also able to call SNVs with high accuracy from Oxford Nanopore long-read data without any modification to the likelihood model. Although the precision and recall was lower than PacBio reads at a similar coverage, this is expected due to the higher error rate of Nanopore reads. Continued improvements in the sequencing technology and the raw basecalling, and the use of context-specific error models for local realignment are expected to further improve the accuracy of variant calling using Nanopore reads.

In this study, we focused on the detection and phasing of SNVs alone, as accurate calling of short indels using SMS reads is challenging due to the high insertion/deletion error rate. A recently developed deep-learning-based variant caller[24] had low precision (0.589) and recall (0.12) for short indel calling on PacBio WGS data. In comparison with CLR reads, PacBio circular consensus sequencing (CCS) produces reads with greater accuracy by sequencing multiple times around the same DNA template. Recent improvements have enabled the generation of highly accurate long reads (10–15 kb read lengths and error rates <1%) using CCS[40]. We expect that using Longshot with these low-error reads will improve the accuracy of SNV calling and also enable accurate short indel calling. As the cost of SMS technologies continues to decrease, these technologies are likely to see widespread use in human disease studies in the near future. In particular, whole-genome SMS can enable the detection of disease-associated SVs and variants in repetitive regions of the genome that cannot be identified using standard Illumina WGS[41–43]. Tools such as LongShot will be valuable for realizing the potential of SMS technologies for the comprehensive detection of all forms of genetic variation in such studies.

## Methods

**Identification of candidate SNVs.** The first step in the Longshot algorithm is to identify positions in the genome that may contain an SNV. Potential SNVs are identified from the pileup of aligned reads by performing a genotype likelihood calculation similar to Samtools or other NGS variant calling methods[3] (see Supplementary Methods). The prior probabilities for genotypes are defined using a slight modification of the approach of Li et al.[44] (see Supplementary Methods). SNV sites for which the posterior probability of a non-reference genotype is >0.01 are considered as candidate SNVs for the next step of the algorithm. The sites are also filtered for minimum read coverage (6 by default), minimum alternate allele count and fraction (3 and 0.125 by default).

**Local realignment using pair-HMMs.** For an SMS read that overlaps a candidate biallelic SNV site with two alleles "ref" and "alt," we want to determine which allele is the most likely observation (allele call) and also assign a probability of error to this observation (quality value). To accomplish this, we perform local realignment of a short sequence from the read to the reference and to the alternate sequence (with the SNV allele added, see Fig. 1b). This local realignment is performed using a pair-HMM[29]. The parameters for the HMM are estimated directly from the aligned reads prior to realignment (see Supplementary Methods).

It is sufficient to perform the local realignment within a short window covering the SNV site. This window is defined using the nearest non-repetitive anchor sequences of length $k$ (default $k = 6$), to the left and right of the SNV where the read sequence matches the reference sequence perfectly (see Supplementary Methods). Once the window $W$ is identified, we use the forward algorithm to calculate $p_{ref} = P(\text{read}(W) \mid \text{ref}(W))$ and $p_{alt} = P(\text{read}(W) \mid \text{alt}(W))$ where read($W$) is the sequence of the read in the window $W$ defined by the two anchors. We select the allele $a_{max} \in \{\text{ref}, \text{alt}\}$ for which the probability is higher, as the observed allele, and use $\text{phred}\left(1 - \frac{p_{a_{max}}}{p_{ref} + p_{alt}}\right)$ as the allele quality score.

When multiple candidate SNVs are located in close proximity, we define the window to include all such SNVs and use a generalization of the calculation described above to determine alleles and estimate base-quality values (see Supplementary Methods). For computational efficiency, a banded version of the forward algorithm is used. This reduces the complexity to $O(mb)$ where $b$ is the width of the band and $m$ is the length of the window (50–200 bp). Allele observations with phred-scaled quality score below a threshold (7.0 by default) are discarded. This reduces the effective read depth for SMS reads (Supplementary Fig. 9).

In order to remove false variants resulting from strand-specific sequencing errors, we filter potential SNVs whose allele observations are over-represented in reads from one strand. For each potential SNV, we build a contingency table of the counts of the reference and alternate allele on reads from the forward strand and reverse strand, respectively. Variants for which the Fisher's exact test $p$-value (two-tailed) is <0.01 are not considered for haplotype-informed genotyping.

**Haplotype-informed genotyping.** Longshot achieves accurate variant calling using SMS reads by performing phased genotyping for all candidate SNVs jointly. Given a set of candidate SNVs $V$ and the allele calls (and quality values) for each read $r \in R$, we aim to maximize the likelihood $p(R|H)$ where $H$ is a pair of haplotypes $(H_1, H_2)$ over the variant set $V$. Longshot optimizes the likelihood function using an iterative approach that uses (i) the HapCUT2 algorithm[13] to estimate the most likely pair of haplotypes for variants with heterozygous genotypes and (ii) local updates to estimate the most likely phased genotype for each variant given the current haplotype pair (Fig. 1c).

Assuming independence between reads, the likelihood function $p(R|H)$ can we written as[13]:

$$p(R|H) = \prod_r p(r|H) = \prod_r \frac{p(r|H_1) + p(r|H_2)}{2}$$

$p(r|H_1)$ for any read $r$ can be calculated using the pair-HMM probabilities for each (read, variant) pair. Let $G$ be the set of possible phased genotypes for a biallelic variant: $\{0|0, 0|1, 1|0, 1|1\}$ (homozygous reference, the two heterozygous states, and homozygous alternate). Let $H$ refer to the current estimate of the most likely haplotype pair and $H^{i,g}$ refer to the haplotype pair $H$ with the $i$th SNV altered to have the phased genotype $g$. Given $H$, we can calculate the posterior probability for the phased genotype $g$ as follows:

$$p(H[i] = g | R, H) = \frac{p(g)p(R|H^{i,g})}{\sum_{g' \in G} p(g')p(R|H^{i,g'})} \quad (1)$$

The optimization starts with the initial set of variants identified from the pileup-based likelihood calculation and the unphased genotypes for each variant estimated using the local realignment. The iterative phase of the Longshot algorithm consists of the following steps:

For $i = 1 \dots k$

1. $L = p(R|H)$
2. Let $V'$ be the set of heterozygous SNVs in $V$
3. $H(V') = \text{HapCUT2}(R, V')$
4. Repeat:
   (a) For each variant $v \in V$: update $H[v]$ using Eq. (1)
   (b) If no genotype was updated in (a), BREAK
5. $L' = p(R|H)$
6. If $\frac{\log(L') - \log(L)}{\log(L)} < \Delta$: BREAK

In Step 3, HapCUT2 is used to phase the current set of heterozygous variants. Then, the haplotype scaffold is used to refine the genotypes of each variant in Step 4. This serves to remove false heterozygous variants and identify new heterozygous variants that can be phased by HapCUT2 in the next iteration. Steps 1–5 are repeated until the relative log-likelihood of the data between consecutive iterations is smaller than $\Delta$ (default $= 1 \times 10^{-5}$).

**Variant filtering.** The raw variant calls were subjected to three types of filters to reduce false positives. SNVs were first filtered according to the GQ estimated by the variant caller. The GQ cutoff was fixed at 50 for short reads. For Longshot, we used a variable GQ cutoff (matched to the median read coverage) for filtering variants. This was done to reduce the number of false SNVs due to true indel variants that have high GQ. For simulations, which do not have indel variants, we used a fixed GQ cutoff of 50.

To filter false SNVs due to copy number amplifications, a maximum read depth filter similar to what has been used previously for short-read-based variant calling was used[45]. Variants with read depth $> d + 5\sqrt{d}$, where $d$ is the median read depth across the entire dataset were filtered out. We also observed that for SMS reads, many false-positive SNVs occur nearby each other in dense clusters. These dense clusters may result from systematically mismapped reads due to missing sequence in the reference genome or are indicative of structural variations such as copy number variants (CNVs). We used a simple density filter (>10 SNVs in a window of 500 bp) to filter out such false variants for variants called with Longshot. For the AJ trio, variants in the Delly exclusion regions (available from https://github.com/tobiasrausch/delly/blob/master/excludeTemplates/human.hg38.excl.tsv) were also filtered out for the analysis of Mendelian consistency.

**Simulations.** We simulated a diploid genome using the reference human genome sequence with heterozygous SNVs (rate = 0.001) and homozygous SNVs (rate = 0.0005) (see Supplementary Methods for details). Paired-end 100 bp reads were generated from the simulated genome with a substitution error rate of 0.001[46]. The short reads were aligned to the human reference (hs37d5) using BWA-MEM and

variants were called using FreeBayes[22]. Similarly, we used SimLoRD[47] to generate PacBio SMS reads (median length = 7.5 kb) from the simulated genome using the default error rates of 0.11 for insertion, 0.04 for deletion, and 0.01 for substitution[47]. The `-mp 1` option was used to force each read to only have a single sequencing pass, so that the error profile of the reads resembles PacBio CLRs (lower accuracy) as opposed to circular consensus reads (greater accuracy). We aligned the SMS reads to the human reference (hs37d5) using the long-read alignment tools BLASR (`v5.3.2`, options `-nproc 16 -bestn 1 -bam`), MiniMap2 (`v2.11-r797`, options `-t 16 -ax map-pb`), BWA-MEM (`v0.7.17`, options `-x pacbio -t 16 -T 0`), and NGMLR (`v0.2.7`, options `-t 16 -x pacbio`).

**Whole-genome sequencing data.** The 45× coverage PacBio SMRT reads for NA12878, aligned to the hs37d5 reference genome using BLASR, were obtained from the GIAB consortium[48]. PacBio read data for the AJ trio was also obtained from the GIAB ftp site and aligned to the hg38 reference genome using BLASR[31], using the same parameters used for aligning the simulated reads. Oxford Nanopore reads for NA12878 were obtained from the Nanopore WGS Consortium[26] and aligned to hg38 using minimap2. Illumina WGS data for NA12878 and the AJ Trio (NA24385, NA24143, and NA24149), sequenced on the HiSeq 2500 (30× and 60× coverage, respectively, 148 bp paired-end reads), was obtained from the GIAB. The 60× coverage datasets were downsampled to half coverage. The reads were downloaded in BAM format aligned to hs37d5 using bwa-mem (NA12878) and hg38 using NovoAlign (AJ trio). Variant calling on Illumina WGS data was performed using FreeBayes[22] (`v1.0.2-33-gdbb6160`) with `-standard-filters` and `-genotype-qualities` turned on). BED files for segmental duplications and repeat elements in the human genome were obtained from the UCSC table browser[49].

**Assessment of variant calling and phasing accuracy.** High-confidence variant call sets generated by the GIAB project were used for assessing accuracy of SNV calling[15,48]. For NA12878, SNVs were compared against the GrCh37 (for Illumina and PacBio) or GrCh38 (for Oxford Nanopore) version of the GIAB high-confidence call set (release v3.3.2). For the AJ trio, SNVs were compared against the GrCh38 version of the GIAB high-confidence call set (release v3.3.2). For comparing the accuracy of Longshot with Clairvoyante and WhatsHap, the GrCh37 version of the calls were used (release v3.3.2). For each individual, the comparison of SNV calls was limited to high-confidence regions (provided in a bed file). Precision and Recall were calculated using RTGtools (`v3.9.1`) vcfeval.

For NA12878, we compared the accuracy of the Longshot haplotypes using the Platinum Genomes haplotypes for the same individual as ground truth. For NA24385, we generated high-quality haplotypes from a consensus of the GIAB trio-based phased genotypes and 10× Genomics phased variant calls and used the resulting haplotypes for assessment of haplotyping accuracy. The haplotypes were compared at all unfiltered SNVs that were called heterozygous in both the assembled haplotypes and the ground truth. The errors were tabulated in terms of the total combined rate of switch and mismatch errors, also known as long-switch and short-switch errors, respectively[13,50]. The N50 metric—defined as the length $N$ in base pairs such that half of the phased portion of the genome is in haplotype blocks of length $N$ or greater—was used to measure the completeness of haplotype blocks.

For the AJ trio, Mendelian consistency of the SNV calls was assessed using RTGtools[51]. For this, SAMtools[27] and BEDtools[52] were used to obtain a set of regions that have high coverage (>20×) of well-mapped SMS reads (*MAPQ* > 30 and filter `-F 3844` applied) in all three individuals. These regions were further intersected with a bed file for the region being investigated (either GIAB confident regions, outside GIAB confident regions, or 95% similar segmental duplications). The individual VCF (variant call format) files for the trio were merged into a single VCF and filtered so that all records have a GQ > 50.

**Server configuration.** All experiments in this study were performed on CentOS 6.6 with Intel Xeon CPU E5-2670 0 @ 2.60 GHz, with jobs managed by a Torque/PBS system.

**Reporting summary.** Further information on research design is available in the Nature Research Reporting Summary linked to this article.

## Data availability
The PacBio and Illumina sequence datasets and variant calls used in this study are publicly available from the GIAB ftp site: ftp://ftp-trace.ncbi.nlm.nih.gov/giab/ftp/. The sub-folders for each individual are as follows:

**NA12878:** data/NA12878/NA12878_PacBio_MtSinai, release/NA12878_HG001/NISTv3.3.2/GRCh37/, data/NA12878/NIST_NA12878_HG001_HiSeq_300x

**NA24385:** data/AshkenazimTrio/HG002_NA24385_son/PacBio_MtSinai_NIST, release/AshkenazimTrio/HG002_NA24385_son/NISTv3.3.2/GRCh38, data/AshkenazimTrio/HG002_NA24385_son/NIST_HiSeq_HG002_Homogeneity-10953946

**NA24149:** data/AshkenazimTrio/HG003_NA24149_father/PacBio_MtSinai_NIST/, release/AshkenazimTrio/HG003_NA24149_father/NISTv3.3.2/GRCh38, data/

AshkenazimTrio/HG003_NA24149_father/NIST_HiSeq_HG003_Homogeneity-12389378

**NA24143:** /data/AshkenazimTrio/HG003_NA24143_mother/PacBio_MtSinai_NIST, release/AshkenazimTrio/HG004_NA24143_mother/NISTv3.3.2/GRCh38, data/AshkenazimTrio/HG004_NA24143_mother/NIST_HiSeq_HG004_Homogeneity-14572558

For the direct comparison between methods, BAMs aligned using NGMLR from the Clairvoyante study were used[34]. The BAMs were obtained from http://www.bio8.cs.hku.hk/clairvoyante/bamUsed/.

The Oxford Nanopore sequence dataset is publicly available from the Nanopore WGS Consortium: https://github.com/nanopore-wgs-consortium/NA12878/blob/master/Genome.md. The NA12878 genome was sequenced using the Oxford Nanopore MinION with version R9.4 of the chemistry on 39 flowcells. We used the rel6 version of the base calls (called using ONT Guppy basecalling software version 2.3.8 + 498297c). All other relevant data are available upon request. The source data underlying Supplementary Fig. 9 is provided as a Source Data file.

## Code availability
Longshot is implemented in the Rust programming language, uses the rust-bio and rust-htslib libraries[53], and the HapCUT2 C code[13]. It is freely available for download at https://github.com/pjedge/longshot. It is also available on Bioconda[54]. A Snakemake workflow[55] for automatically generating all of the results of the paper is available at https://github.com/pjedge/longshot_study.

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

## Acknowledgements

The research was supported by the National Human Genome Research Institute of the National Institute of Health (award number R01HG010149).

## Author contributions

P.E. designed and implemented the algorithm, performed the analyses, and wrote the manuscript. V.B. conceived the project, designed the algorithm, performed analyses, and wrote the manuscript.

## Competing interests

The authors declare no competing interests.
