## [Peer Review File · Nature Communications]

Reviewers' comments:

Reviewer #1 (Remarks to the Author):

The author introduces a new variant calling method, Longshot, for calling SNVs from single-molecule sequencing (SMS) reads. Longshot achieves higher SNV calling precision and recall than other SMS variant callers, in both Pacbio reads and Oxford Nanopore reads. Compared to variant calling using Illumina reads, Longshot (using SMS reads) performs well especially in repetitive regions (e.g. segmental duplications). Besides variant calling, Longshot also phases the variants into long haplotype blocks with high accuracy and completeness (compared to using Illumina reads alone).

While these results may advance the use of SMS technologies in various scenarios, I have concern about the novelty of the method.

(A) The method has 3 major steps. Step 1 is candidate variant identification, which uses the standard pileup approach to determine which sites are potential variants. Step 2 is allelotyping via local realignment. In this step, each read (fragment) is being classified as coming from the REF sequence or the ALT sequence, by realigning the read fragment to both sequences using HMM forward algorithm. If there are multiple potential variants nearby, they will be considered together, and an exponential number of ALT sequences will be generated for realignment. For example, if 3 nearby potential variants are clustered, then all $2^3=8$ haplotype combinations will be compared to pick out the best one. This might not be a standard technique, but it can be found in other similar tools, such as Nanopolish (a HMM-based Nanopore variant caller). Although the objective of Nanopolish is not to find long haplotypes, it can as well output the per-read "local haplotype" in a short window just like Step 2 of Longshot.

(B) In my opinion, Step 3 of Longshot is what differentiates Longshot from Nanopolish. In Step 3, the called alleles (and quality values) at the potential variant sites for each read are taken as input. An existing haplotype assembly tool HapCUT2 is used to estimate the most probably haplotypes over a long region. The estimated haplotypes are then further refined to maximize likelihood. The authors compared the haplotypes obtained using this 3-step Longshot workflow to those using HapCUT2 alone (given a raw list of variants found using Illumina reads), and concluded that their accuracy and completeness were very similar. This makes me wonder if the power of Longshot in fact comes from HapCUT2. It would be interesting to see what happens if the first 2 steps of Longshot is replaced by other SMS variant callers (plus some simple methods to assign called alleles to individual reads).

Other minor concerns and issues:

(1) In Table S7, Longshot is compared to other SMS variant callers. However, the benchmark figures of all variant callers except Longshot were obtained from paper preprints, not from benchmarking runs by the authors. To my knowledge, these variant callers used a variety of mapping tools when doing their own benchmarks -- e.g., some chose Minimap2 over BLASR for speed concern. It is not entirely fair to compare a variant caller with Minimap2 against another caller with BLASR. (According to Figure S2, Longshot+Minimap2 indeed had worse recall than Longshot+BLASR.) Moreover, it is not uncommon that evaluation methods in different papers (even using the same dataset) have small discrepancies due to thresholding, filtering, or other customizations. It would be great if the authors can evaluate the existing tools again on their own, giving a more up-to-date and fair benchmark comparison.

(2) One of the major claims in the paper is that Longshot (and SMS sequencing) has excellent recall rate in repetitive regions (e.g. segmental duplications) compared to Illumina-based methods. According to Figure S2 (bottom right), the excellent recall rate in fact relies very much on the mapping tool used. This suggests that the SMS read mapping tool plays a much bigger role than the variant caller in repetitive regions -- which makes sense. It would be interesting to include a comparison of Longshot against other SMS variant callers in repetitive regions.

(3) It is not obvious whether the run time in Table S1 refers to wall-clock hours or CPU hours. Main memory requirement (which can be a headache for other variant callers) and server configuration were not mentioned either.

(4) The iterative genotype-haplotype algorithm on page 8 is not clear enough. In each iteration, H gets updated, and the likelihood L is recomputed. In the next iteration, H gets reassigned as HapCUT2(R,V), but neither R nor V was updated in the previous iteration. I assume that there is some hidden parameter to HapCUT2() which is modified along with H. The hidden parameter might be H itself.

Experiments are well-designed and appropriate in general. They are also reproducible as the authors provided the relevant workflow scripts.

Reviewer #2 (Remarks to the Author):

Review of Edge and Bansal, Longshot: accurate variant calling in diploid genomes using single-molecule long read sequencing.

The authors present a method to detect SNVs using long-read sequences. The method makes sense, although some of the methods that incorporate information from multiple reads (e.g. the pacbio consensus tools) may outperform this for recall. The results demonstrate that, when combined with filtering on variants near indels, have only about twice the false positive rate of SNV calls versus Illumina. This is close to feasible for precision medicine/Mendelian studies. The section mentioning Longshot's performance versus machine learning methods sells the results short. For disease studies, swapping precision with false discovery rate is a more informative metric for two reasons: one will look for genes mutated in multiple individuals, or one tests (Sanger) all the variants in one individual. The performance of longshot is actually **considerably** better than the other methods when considering FDR, and this should be highlighted.

The software runs quite easily. It will increase use by the community if multithreading is enabled.

The major suggestion is to compare against the quiver method produced by PacBio, which can also perform variant calling. It does not perform well on heterozygous variants, and the expectation is that longshot will outperform the PB method for this type of variant. It may be a little worse for homozygous variants, but those are less frequent. The release of the consensus methods via bioconda will help make this analysis feasible, whereas before it was pretty much impossible to install. The authors state that quality values are less meaningful for SMS reads than Illumina, however they are used in the PacBio consensus tools.

Also, it would be useful to compare the Longshot SNV + WhatsHap phasing, since this tool is quite popular for long read phasing, although the results will likely not be significantly different.

There are some minor suggestions, in no particular order.

Figure 2 is nice to see, but not terribly informative and should probably be a supplementary figure. However, this can be at the discretion of the authors.

The Nanopore analysis should be careful to note the exact date of the sequencing and the version of the basecaller used on the data -- ideally even the parameters of the basecaller. All of these have considerable effect on variant calling.

State in the text the FDR of Longshot versus the ML methods.

List how many GIAB "high-confidence" calls are not supported by Longshot. Some of the high confidence calls will be false positives, and it will be help for the reference variant panel community to know which ones. This will probably just be a few thousand, but they are aiming at 0 false positives, so this will help.

How many coding bases are you able to call variants on with longshot in segdup regions (e.g. potential for precision medicine)?

It is worth discussing the gap between Illumina analysis for disease studies, and what Longshot enables in the discussion.

Response to reviewers' comments:

Reviewer #1 (Remarks to the Author):

The author introduces a new variant calling method, Longshot, for calling SNVs from single-molecule sequencing (SMS) reads. Longshot achieves higher SNV calling precision and recall than other SMS variant callers, in both Pacbio reads and Oxford Nanopore reads. Compared to variant calling using Illumina reads, Longshot (using SMS reads) performs well especially in repetitive regions (e.g. segmental duplications). Besides variant calling, Longshot also phases the variants into long haplotype blocks with high accuracy and completeness (compared to using Illumina reads alone).

While these results may advance the use of SMS technologies in various scenarios, I have concern about the novelty of the method.

(A) The method has 3 major steps. Step 1 is candidate variant identification, which uses the standard pileup approach to determine which sites are potential variants. Step 2 is allelotyping via local realignment. In this step, each read (fragment) is being classified as coming from the REF sequence or the ALT sequence, by realigning the read fragment to both sequences using HMM forward algorithm. If there are multiple potential variants nearby, they will be considered together, and an exponential number of ALT sequences will be generated for realignment. For example, if 3 nearby potential variants are clustered, then all $2^3=8$ haplotype combinations will be compared to pick out the best one. This might not be a standard technique, but it can be found in other similar tools, such as Nanopolish (a HMM-based Nanopore variant caller). Although the objective of Nanopolish is not to find long haplotypes, it can as well output the per-read "local haplotype" in a short window just like Step 2 of Longshot.

(B) In my opinion, Step 3 of Longshot is what differentiates Longshot from Nanopolish. In Step 3, the called alleles (and quality values) at the potential variant sites for each read are taken as input. An existing haplotype assembly tool HapCUT2 is used to estimate the most probably haplotypes over a long region. The estimated haplotypes are then further refined to maximize likelihood. The authors compared the haplotypes obtained using this 3-step Longshot workflow to those using HapCUT2 alone (given a raw list of variants found using Illumina reads), and concluded that their accuracy and completeness were very similar. This makes me wonder if the power of Longshot in fact comes from HapCUT2. It would be interesting to see what happens if the first 2 steps of Longshot is replaced by other SMS variant callers (plus some simple methods to assign called alleles to individual reads).

Most variant callers for high-throughput sequence data follow a similar paradigm: identify candidate alleles at a locus, allelotype the reads, and call variant sites and genotypes. As observed by the reviewer, the key novelty of our method LongShot lies in the third step. Unlike variant callers for Illumina data or deep-learning based variant callers, LongShot attempts to estimate phased genotypes across multiple variants sites jointly using a maximum likelihood

model. To demonstrate the value of the phased genotyping (using HapCUT2) and other components of LongShot for variant calling accuracy, we have added a figure (Supplementary Figure 6) where we show the precision-recall curves for SNV calling comparing the full Longshot algorithm and the algorithm without phased genotyping (skipping step 3 of the algorithm). We have add the following text in the results:

“We investigated the importance of the phased genotyping for the accuracy of Longshot by running it on the NA12878 PacBio dataset (downsampled to 30x coverage) without phased genotyping (essentially skipping step 3 of the algorithm). We found that skipping the phased genotyping reduced Longshot's recall significantly from 0.959 to 0.905 (genotype quality threshold of 30) while the precision remained virtually unchanged (0.994). Therefore, the HapCUT2-based phased genotyping step is indeed an important contributor to the accuracy of the LongShot algorithm”.

For comparison, we also used the "WhatsHap Genotype" method to genotype the same data using the potential SNVs identified in step 1 of the Longshot algorithm and with Longshot's read depth and SNV density filters applied. We found that the precision and recall using Whatshap were lower than the those with Longshot (precision=0.984 and recall=0.952).

Other minor concerns and issues:

(1) In Table S7, Longshot is compared to other SMS variant callers. However, the benchmark figures of all variant callers except Longshot were obtained from paper preprints, not from benchmarking runs by the authors. To my knowledge, these variant callers used a variety of mapping tools when doing their own benchmarks -- e.g., some chose Minimap2 over BLASR for speed concern. It is not entirely fair to compare a variant caller with Minimap2 against another caller with BLASR. (According to Figure S2, Longshot+Minimap2 indeed had worse recall than Longshot+BLASR.) Moreover, it is not uncommon that evaluation methods in different papers (even using the same dataset) have small discrepancies due to thresholding, filtering, or other customizations. It would be great if the authors can evaluate the existing tools again on their own, giving a more up-to-date and fair benchmark comparison.

We agree that a fair comparison should be based on evaluating each tool using the same set of aligned BAMs. However, we were unable to successfully call variants using Clairvoyante on the BLASR-aligned BAMs that we used in our study. We communicated with the developer of Clairvoyante who informed us that they don't have a BLASR model for Clairvoyante and BLASR performs worse than NGMLR and minimap2 for their method. In the Clairvoyante paper, the authors used datasets that have been aligned with the NGMLR tool and have provided trained models using the same alignment tool. Therefore, we have added a new direct comparison of Longshot, WhatsHap and Clairvoyante by running these tools on the NGMLR-aligned BAMs obtained from the Clairvoyante paper (table 1). For WhatsHap, the tool doesn't support end-to-end variant calling but only 'genotyping', therefore, we used the candidate variants identified by LongShot (Step 1) as input to WhatsHap. The second paragraph on page 5

describes the comparison of LongShot with the other tools (summary in Table 1). We find that LongShot had significantly better precision as well as recall than Clairvoyante and WhatsHap on multiple PacBio sequenced genomes.

For Deepvariant, we have updated Supp Table 4 to show the accuracy on three chromosomes from the NA12878 PacBio BLASR-aligned bams that were used in our study. In the DeepVariant paper, the remaining 19 chromosomes were used for training and therefore, comparison of the precision/recall values on the 3 chromosomes represents a direct comparison. We decided not to benchmark against Deepvariant on other datasets because the open-source version of Deepvariant does not support CLR reads (as communicated by the authors of DeepVariant: <https://github.com/google/deepvariant/issues/174>).

(2) One of the major claims in the paper is that Longshot (and SMS sequencing) has excellent recall rate in repetitive regions (e.g. segmental duplications) compared to Illumina-based methods. According to Figure S2 (bottom right), the excellent recall rate in fact relies very much on the mapping tool used. This suggests that the SMS read mapping tool plays a much bigger role than the variant caller in repetitive regions -- which makes sense. It would be interesting to include a comparison of Longshot against other SMS variant callers in repetitive regions.

We agree that this would be a useful comparison, however, as mentioned in the previous comment, we were unable to run Clairvoyante on the BLASR aligned bams and were also unsuccessful in running DeepVariant due to the lack of support for CLR data. NGMLR performed poorly for variant calling in segmental duplications on simulated data (Figure S2) and therefore, we did not analyze the NGMLR-based variant calls in segmental duplications.

(3) It is not obvious whether the run time in Table S1 refers to wall-clock hours or CPU hours. Main memory requirement (which can be a headache for other variant callers) and server configuration were not mentioned either.

We have specified that the runtime is the total wallclock hours over all chromosomes, using a single core. We have also described the CPU configuration in more detail in the caption of Table S1, and described the server configuration in more detail in the methods section.

(4) The iterative genotype-haplotype algorithm on page 8 is not clear enough. In each iteration, H gets updated, and the likelihood L is recomputed. In the next iteration, H gets reassigned as HapCUT2(R,V), but neither R nor V was updated in the previous iteration. I assume that there is some hidden parameter to HapCUT2() which is modified along with H. The hidden parameter might be H itself.

The hidden parameter that changes between iterations is V', the set of heterozygous variants. V' changes in step 4 when the individual genotypes H[v] are updated. We have updated

pseudocode (line 3) to make it clear that HapCUT2 only updates the haplotypes for the heterozygous variants: $H(V) = \text{HapCUT2}(R, V)$.

We have also added additional text after the pseudocode to make this more clear: "In Step 3, HapCUT2 is used to phase the current set of heterozygous variants. Then, the haplotype scaffold is used to refine the genotypes of each variant in the loop in step 4. This serves to remove false heterozygous variants and identify new heterozygous variants that can be phased by HapCUT2 in the next iteration."

Experiments are well-designed and appropriate in general. They are also reproducible as the authors provided the relevant workflow scripts.

We thank the reviewer for appreciating our work.

Reviewer #2 (Remarks to the Author):

The authors present a method to detect SNVs using long-read sequences. The method makes sense, although some of the methods that incorporate information from multiple reads (e.g. the pacbio consensus tools) may outperform this for recall. The results demonstrate that, when combined with filtering on variants near indels, have only about twice the false positive rate of SNV calls versus Illumina. This is close to feasible for precision medicine/Mendelian studies. The section mentioning Longshot's performance versus machine learning methods sells the results short. For disease studies, swapping precision with false discovery rate is a more informative metric for two reasons: one will look for genes mutated in multiple individuals, or one tests (Sanger) all the variants in one individual. The performance of longshot is actually ****considerably**** better than the other methods when considering FDR, and this should be highlighted.

We thank the reviewer for the positive feedback and recognizing the high precision of our method. We have updated the text to highlight the low FDR of our method:

Results: "In particular, Longshot achieved very high precision or a low false discovery rate (FDR) of 0.5%. In comparison, the FDR for Clairvoyante was 3-fold higher, 1.6%."

Discussion: "Our method has a very low false discovery rate (0.5-0.8%) across multiple whole-genome PacBio datasets that is 2-4 fold lower than other variant calling methods. Furthermore, we find that the FDR can be reduced further to 0.3% by filtering out known common indels."

The software runs quite easily. It will increase use by the community if multithreading is enabled.

The software can be parallelized at the process level by running on individual chromosomes or even sub-chromosomal chunks. We plan to release a script that will automate this process.

The major suggestion is to compare against the quiver method produced by PacBio, which can also perform variant calling. It does not perform well on heterozygous variants, and the expectation is that longshot will outperform the PB method for this type of variant. It may be a little worse for homozygous variants, but those are less frequent. The release of the consensus methods via bioconda will help make this analysis feasible, whereas before it was pretty much impossible to install. The authors state that quality values are less meaningful for SMS reads than Illumina, however they are used in the PacBio consensus tools.

We installed Genomicconsensus 2.3.3 from bioconda which supports both Arrow and Quiver. We attempted to use Quiver to call variants for the NA24385 genome (the tool does not support the NA12878 dataset's older BAM format, and the more recent algorithm Arrow does not support the AJ trio's older P5-C3 chemistry). The tool crashed on several of the chromosomes with a cryptic error message that has been described previously in pbbioconda's github issues (<https://github.com/PacificBiosciences/pbbioconda/issues/130>) but not resolved, therefore, we were unable to compare against Quiver.

Also, it would be useful to compare the Longshot SNV + WhatsHap phasing, since this tool is quite popular for long read phasing, although the results will likely not be significantly different.

We have added the results from running WhatsHap to Supp. Fig S8. The switch and mismatch error rate for WhatsHap phasing were comparable to those using Longshot and HapCUT2. Both LongShot and HapCUT2 provide a phasing quality (PQ) value and the pruned haplotypes (PQ > 30) had lower switch and mismatch error than those assembled using WhatsHap. WhatsHap does not provide a confidence metric for its phased SNVs.

There are some minor suggestions, in no particular order.

Figure 2 is nice to see, but not terribly informative and should probably be a supplementary figure. However, this can be at the discretion of the authors.

We have removed Figure 2 from the main text. The extended version of Figure 2 is already in the supplement and contains the same results as well as the results for other variant callers. We have changed the order of the paragraphs in that section to make it consistent with this change.

The Nanopore analysis should be careful to note the exact date of the sequencing and the version of the basecaller used on the data -- ideally even the parameters of the basecaller. All of these have considerable effect on variant calling.

We have added the ranges of the dates of sequencing, the version of Guppy, and what is known of the Guppy parameters to the data availability section.

State in the text the FDR of Longshot versus the ML methods.

We have added the following line to the results: “In particular, Longshot achieved very high precision or a low false discovery rate (FDR) of 0.5%. In comparison, the FDR for Clairvoyante was 3-fold higher, 1.6%.”

List how many GIAB “high-confidence” calls are not supported by Longshot. Some of the high confidence calls will be false positives, and it will be help for the reference variant panel community to know which ones. This will probably just be a few thousand, but they are aiming at 0 false positives, so this will help.

The false negative rate of LongShot is 3-4%, so we cannot say with high confidence that any GIAB variant that is not supported by Longshot is a false positive. However, we can identify potential false negatives in GIAB data using LongShot variant calls that don’t overlap indels. We find 5903 variants called by Longshot inside GIAB confident regions that are not within 5 bp of any GIAB variant, and have added this finding to the Discussion.

“Furthermore, some of the false positives calls by LongShot may actually correspond to false negatives in the GIAB high-confidence call sets. A recent graph-based read alignment approach identified several thousand variants that were absent in the GIAB call-sets. Many of these variants are located in repetitive and variant-dense genomic regions that are problematic for mapping using short reads but should be callable using long single-molecule reads. In the NA12878 genome, LongShot identified ~5900 SNVs that are located in GIAB high-confidence regions and do not overlap indels present in the GIAB variant calls. Further analysis of these variants will be helpful in improving the recall of gold-standard variant call-sets for human genomes.”

How many coding bases are you able to call variants on with longshot in segdup regions (e.g. potential for precision medicine)?

*We have added this analysis to the section “Analysis of SNV calls in repetitive regions”:
“We found that in total, 1.66 Mb of the bases in segmental duplications with $\geq 95\%$ similarity overlap with coding exons and 90.3% of these bases were well-mapped in the 45x PacBio dataset (each position having at least 20x coverage and $\geq 90\%$ of reads aligned to the position having MAPQ ≥ 30).”*

It is worth discussing the gap between Illumina analysis for disease studies, and what Longshot enables in the discussion.

We have added the following text to the discussion: “As the cost of SMS technologies continues to decrease, these technologies are likely to see widespread use in human disease studies in the near future. In particular, whole-genome SMS can enable the detection of

disease-associated structural variants and variants in repetitive regions of the genome that cannot be identified using standard Illumina WGS [41,42,43]. Tools such as LongShot will be valuable for realizing the potential of SMS technologies for the comprehensive detection of all forms of genetic variation in such studies.”

Reviewers' comments:

Reviewer #1 (Remarks to the Author):

The authors have addressed most of my concerns, with the follows remaining:

1. It is necessary also to benchmark "Nanopolish" as it is the de facto standard for variant calling using ONT reads.
2. In the newly added Table 1, Longshot and WhatsHap's run times are proportionate to the read coverage of the genome, but Clairvoyant is not. We had a problem running Clairvoyant but later sped up by using bioconda automatic environment configuration. We suggest the authors double check if all tools are properly configured and are using up the number of cores assigned to them.

Reviewer #2 (Remarks to the Author):

The authors have thoroughly addressed my comments.

Reviewer #1 (Remarks to the Author):

The authors have addressed most of my concerns, with the follows remaining:

1. It is necessary also to benchmark "Nanopolish" as it is the de facto standard for variant calling using ONT reads.

Response: *We are thankful to the reviewer for this suggestion. We have benchmarked LongShot and Nanopolish using the latest release of the Oxford Nanopore data (rel6) and the results demonstrate that Longshot achieves high accuracy (precision = 0.99 and recall = 0.933) on ONT data that is much better than Nanopolish. We used the rel6 version of the basecalls for variant calling since it incorporates the latest basecalling improvements and also uses the multifast5 format for storing the raw signal files which are needed for running Nanopolish.*

We have updated the section on "SNV calling using Oxford Nanopore reads" as follows:

"On the latest version of this ONTdataset, LongShot achieved a precision equal to 0.991 and recall value equal to 0.933 at a GQ threshold of 65 for SNV calling (see Supplementary Figure 8 for a precision-recall curve). For comparison, we called variants using Nanopolish, a software tool for signal-level analysis of Oxford Nanopore data (36). Nanopolish required more than 43 hours to call variants on chromosome 20 using 4 cores and achieved a best F1 score of 0.93 (Supplementary Figure 8). Longshot comprehensively outperformed Nanopolish with a best F1 score of 0.967 and took only 5 hours and 13 minutes for variant calling (using a single core)."

The precision-recall curve in Supplementary Figure 8 has been updated to include a comparison of LongShot and Nanopolish on chromosome 20.

Since the accuracy of LongShot on ONT data improved significantly using the new version of the dataset, we have also updated the discussion appropriately (highlighted in blue in the text).

2. In the newly added Table 1, Longshot and WhatsHap's run times are proportionate to the read coverage of the genome, but Clairvoyant is not. We had a problem running Clairvoyant but later sped up by using bioconda automatic environment configuration. We suggest the authors double check if all tools are properly configured and are using up the number of cores assigned to them.

Response: *To assess if Clairvoyante is using the number of assigned cores correctly, we reran it using a single core and the run-times were as follows:*

Genome cov single-core 4-cores

NA12878 44x 49:47 21:44

NA24385 63x 50:02 22:25

NA24385 27x 56:28 21:09

NA24149 26x 57:47 21:42

NA24143 23x 71:43 23:59

The run times using a single-core are 2-3 fold greater than the original run-times (21-23 hours using four cores) and indicate that Clairvoyante is utilizing the assigned number of cores. In addition, the NA24143 dataset had the longest run-time using a single core which was also the case using four cores. This provides further evidence that we have configured this tool correctly. The observed runtimes for Clairvoyante not being proportional to the read coverage may be due to the fact that this tool processes the aligned sequence data differently compared to LongShot and WhatsHap.

Reviewer #2 (Remarks to the Author):

The authors have thoroughly addressed my comments.

REVIEWERS' COMMENTS:

Reviewer #1 (Remarks to the Author):

The authors have addressed all my concerns.